# Policy Evaluation of Demonstration Cooperative Construction: Evidence from Sichuan Province, China

**DOI:** 10.3390/ijerph191912259

**Published:** 2022-09-27

**Authors:** Rui Chen, Nawab Khan, Shemei Zhang

**Affiliations:** College of Management, Sichuan Agricultural University, Chengdu 611130, China

**Keywords:** farmers cooperatives, demonstration cooperatives, policy evaluation, propensity score matching, sustainable rural development

## Abstract

Agricultural cooperatives are professional organizations that increase farmers’ incomes through market failure corrections, livelihood resilience, and sustainable rural development. The main purpose of this paper was to evaluate the policy effects of the construction of demonstration cooperatives for farmers in China. The authors of this paper used the propensity score matching (PSM) method to evaluate the policy effect of the construction of demonstration cooperatives based on questionnaire survey data on 509 farmer cooperatives in 10 counties in Sichuan Province of China. On this basis, the inverse probability weighting-regression adjustment (IPWRA) method was used as a robustness test. The authors of this study were the first to systematically and comprehensively assess the policy effects of demonstration cooperatives while considering selectivity effects. The empirical results show that the Chinese government’s construction of demonstration cooperatives has significant policy effects, especially regarding policy support in improving the economic strength, service capacity, product quality, and social response of demonstration cooperatives. However, the policy effect of improving the democratic management of cooperatives is not significant. It is recommended that the government continue to strengthen policy support in improving the economic strength, service capacity, product quality, and social response of demonstration cooperatives. Simultaneously, more effective measures should be taken to promote the democratic management of model cooperatives.

## 1. Introduction

Agricultural cooperatives are producer-owned and skillful organizations that expand the incomes of farmers by correcting market failures. They organize smallholder farmers to improve farmers’ negotiating power in agricultural production materials and agricultural product markets and promote farmers’ technology adoption and credit access, thereby improving smallholder farmers’ agricultural productivity and technical efficiency [1,2,3,4]. Therefore, the development of agricultural cooperatives has become an important means for developing countries to promote agricultural modernization. The Chinese government also has been investing economic resources in agricultural cooperatives. Since the official implementation of the Law of the People’s Republic of China on Farmers’ Professional Cooperatives in 2007, China’s agricultural cooperatives have rapidly developed. By the end of April 2021, the number of agricultural cooperatives nationwide reached 2.259 million, an 86-fold increase from 26,000 in 2007. The quality improvement of agricultural cooperatives has always been one of the key issues plaguing China’s agricultural development. A consensus has been reached: the number of farmer cooperatives in China is rapidly growing, but the quality of their operations is generally not high [5,6,7]. The development of cooperatives has problems such as small operation scale, irregular internal governance, weak market competitiveness, and deviation from policy orientation [8,9,10]. At present, the growth rate of cooperatives is slowing down. Starting from the implementation of the cooperative law, the annual growth rate of cooperatives dropped from 92.6% in 2012 to 40.9% in 2020, and the growth rate has since narrowed to 1.8% year-on-year. Quantitative growth shifts to quality improvement [11]. Therefore, improving the management quality of agricultural cooperatives, especially the management quality of most cooperatives in the region, is a major problem facing the development of cooperatives at present.

The construction of demonstration cooperatives has always been regarded as an important means to regulate the development and improve the quality of cooperatives [12]. Since 2009, China has been constructing demonstration cooperatives. In 2009, the Ministry of Agriculture, together with 11 departments including the National Development and Reform Commission and the Ministry of Science and Technology, jointly issued the “Opinions on the Construction of demonstration cooperatives of Farmers’ Professional Cooperatives”. The opinion proposed that demonstration cooperatives should be built into modern agricultural management organizations that lead farmers to participate in domestic and foreign market competition. By 2013, the inter-ministerial joint meeting system for the development of national farmer cooperatives was established, and four batches of national demonstration cooperatives were selected in 2014, 2016, 2018, and 2020; a total of 8514 enterprises were identified (Data source: Arranged according to the list of four national farmer cooperative demonstration cooperatives, Nong Jing Fa [2014] No. 10, Nong Jing Fa [2016] No. 16, Nong Jing Fa [2019] No. 2, and Nong Jing Fa [2021] No. 1). The selection criteria of the National Demonstration Cooperatives require members to contribute more than one million yuan, an operating income of more than 4 million yuan in the eastern region. Strict thresholds make a group of cooperatives with good development quality stand out and become industry benchmarks. At present, demonstration cooperatives are divided into four levels (the national, provincial, municipal, and county levels), and the selection criteria are lowered in turn. By November 2021, there were about 168,000 demonstration cooperatives at all levels, accounting for about 7% of the total national cooperatives (Data source: Website of the Central People’s Government of the People’s Republic of China). Compared with other policies, this policy clarifies that relying on the four-level platforms of the ministry, the province, the city, and the county requires the selection of the best to cultivate and support an agricultural cooperative group with large operation scale, strong serviceability, excellent product quality, and democratic management. Demonstration cooperatives enjoy preferential policy support in the distribution of financial funds, credit guarantee loans, agricultural project construction, land transfer, and cooperative talent training. The policy also requires demonstration cooperatives to develop and strengthen themselves while driving the development of other cooperatives. Therefore, the policy has obvious influence in developing and expanding individual cooperatives while promoting the quality improvement of regional cooperatives by demonstration.

According to existing research, policy effects are common in many fields, such as the policy effect on consumer behavior, the policy effect on technological innovation, and the policy effect on the construction of various demonstration areas [13,14,15]. The demonstration mechanism is also a core mechanism of Chinese-style policy implementation. Through repeated experiments under uncertain circumstances to adapt to environmental changes, the experience is extended to resolve the contradictory relationship between “points” and “surfaces” [16,17]. Macro data show that although the number of demonstration cooperatives, especially the national demonstration cooperatives, is not large, their economic strength is strong and their effect of driving farmers is remarkable, becoming a “benchmark” and “model” for other cooperatives to learn from. Many studies have also confirmed that, compared with non-demonstration cooperatives, demonstration cooperatives have more strengthened internal governance, member income increase, and industrial development [18,19]. The newly implemented policy to improve the quality of cooperatives throughout the county also clearly states that “the demonstration cooperatives should be the focus of policy support, and the leading of the demonstration cooperatives should be strengthened”. It can be foreseen that with the continuous enhancement of the individual strength of demonstration cooperatives, the development of demonstration cooperatives will inevitably promote the overall quality improvement of cooperatives.

However, other studies have shown that there is a gap between the development reality of demonstration cooperatives and policy expectations. In the evaluation process of demonstration cooperatives, there are practical standards that deviate from the document standards, e.g., lowering evaluation standards and selectively complying with the document standards [20]. “Unintended consequences” have appeared in the development of demonstration cooperatives, and the development of demonstration cooperatives “has an exaggerated reputation” that has brought about distortions in the image of demonstration cooperatives and unfairness between demonstration and non-demonstration cooperatives [21]. The excessive pursuit of economies of scale in grain demonstration cooperatives has led to insufficient grain production function and hindered the general increase in farmers’ income [22]. In practice, we have also observed that some demonstration cooperatives have gradually fallen from the altar of demonstration, and some of the cooperatives that have been cleaned up through the special clean-up operation of empty shell cooperatives were first demonstration cooperatives. So, is the individual strength of demonstration cooperatives stronger than that of non-demonstration cooperatives? Do demonstration cooperatives have potential to enact policy effects? These questions need to be resolved. Therefore, the main purpose of this paper was to assess the policy effect of demonstration cooperatives. We selected Sichuan Province, where China’s cooperatives are well-developed, as our research area. Based on the sample survey data of cooperatives in 10 counties in Sichuan Province, we uses policy effect evaluation models such as PSM to carry out the research in this paper.

This paper is organized as follows. The analytical framework is presented in Section 2. We present the data, estimation technique, and descriptive statistics in Section 3. The empirical results and discussion are presented in Section 4, and the Section 5 is the conclusion.

## 2. Framework of the Study

### 2.1. Nudge–Imitation Theory

Thaler, winner of the 2017 Nobel Prize in Economics, and his collaborator Sunstein proposed the nudge theory based on the “social person” hypothesis of behavioral economics, arguing that nudge means that no option is prohibited or no significant change is made [23]. Economic incentives are a system of choice that predictably changes people’s behavior. Under the premise that negative effects can be minimized and easily avoided, any factor that can significantly change the behavior of a social person can be called a nudge [23]. A nudge is a small intervention strategy applied by a nudge person to the environment to change the behavioral decision making of the nudged person in order to achieve the expected goal of promoting the healthy development of individuals and society [24]. This kind of intervention strategy is open and transparent, fully guaranteeing the freedom of choice of the nudged and not depending on changing material or immaterial incentives. The government can use “libertarian paternalism” to influence people’s behavioral choices without adopting any prohibition or obvious economic incentives in the process of policy implementation so that they can freely and expectedly choose, before finally realizing the government’s and the people’s incentives [12].

Therefore, in the development of cooperatives in China, the construction of demonstration cooperatives by the government can be regarded as a typical nudge. From the review of the construction and development process of the above-mentioned demonstration cooperatives, it can be found that the government’s policy support of the development of cooperatives is inclined toward demonstration cooperatives, a policy tool used to consciously influence and change the self-selection behavior of farmer cooperatives. Through this “libertarian paternalism”, rather than compulsory administrative intervention, those cooperatives that want to obtain government support will often anchor their decision-making behavior to the standards of demonstration cooperatives and accordingly adjust their behavior. This operation and management method can help a cooperative enter the standardized development track [12].

Tarde, a famous French sociologist, was the first to study imitation. He believed that imitation is innate, a human biological characteristic, and a “basic social phenomenon”, and he proposed three laws of imitation to explain the formation of communities [25], the evolution of human behavior, and the characteristics of information dissemination: first is the internal logic law, that is, the lower class of society tends to imitate the upper class; the second is the law of geometric progression, i.e., in the absence of interference, once the imitation begins, the geometric progression will rapidly grow and spread; and the third is the internal and external law, i.e., the individual’s imitation and selection of local culture and its behavior always take precedence over foreign culture and its behavior [25]. Inter-organizational imitation is a very common way of organizational behavior. Whether it is the adoption of new products or new technologies, the introduction of new management methods or organizational structures, the entry of new markets, or the selection of investment opportunities, there are organizations via the phenomenon of imitating each other [26]. Inter-organizational imitation behaviors tend to imitate those behaviors that represent optimal outcomes [27]. Cooperatives can not only imitate the strategic behaviors or organizational structures of demonstration cooperatives through “formal contractual learning”, acquire knowledge of demonstration cooperatives, and identify new business opportunities but also learn from the development of demonstration cooperatives through “informal alternative learning” (i.e., practical experience) to achieve the desired effect [12]. According to nudge–imitation theory, verifying that demonstration cooperatives have a policy effect requires proof that demonstration cooperatives have developed better than the non-demonstration cooperatives, so it is necessary to judge whether demonstration cooperatives are simply listed or have the strength required to be a model for other cooperatives.

### 2.2. Research Hypothesis

According to document No. 5 of 2019 of the Ministry of Agriculture and Rural Affairs of the People’s Republic of China “Interim Measures for the Evaluation and Monitoring of National Farmers’ Professional Cooperative Demonstration Cooperatives” and document No. 8 of 2010 of the Ministry of Agriculture and Rural Affairs of the People’s Republic of China “Standards for the Establishment of Farmers’ Professional Cooperative Demonstration Cooperatives (Trial)”, demonstration cooperatives should have outstanding achievements in democratic management, business scale, service capacity, product quality, and social reputation, which comprise the foundation for demonstration cooperatives to effectively play their demonstration role. In practice, the China government’s evaluation standards for demonstration cooperatives at all levels often refer to the national standards for demonstration cooperatives and adjust them according to the actual situation of the region, but the evaluation standards have presented convergence. Taking Sichuan as an example, according to document No. 118 of the Sichuan Provincial Department of Agriculture in 2021, the “measures for the evaluation and monitoring of provincial demonstration cooperatives of Sichuan farmers cooperatives” have been registered and established by law, and these cooperatives have implemented democratic management, implemented standardized financial management, demonstrated strong economic strength, and demonstrated obvious service results. The seven aspects of excellent product (service) quality and good social reputation stipulate the evaluation standards of provincial demonstration cooperatives. According to the relevant government evaluation standards and combined with the research needs, the authors of this paper chose to consider the policy evaluation of demonstration cooperatives from five respects: democratic management, economic strength, serviceability, product quality, and social response.

The democratic management of cooperatives means that whether or not members contribute capital or have the status of directors and supervisors, they have equal power in the cooperative, and the decision making and voting of cooperative affairs are based on the primes of one person, one vote. As one of the essential stipulations of cooperatives, democratic management has been advocated by the international cooperative community [28]. However, with the alienation of cooperative members and the increasing role of capital factors in cooperative operation, the principle of the democratic management of cooperatives is facing unprecedented challenges [29]. The prescriptive drift of the nature of the democratic control of cooperatives based on one person, one vote is more obvious in China [30]. Therefore, it is challenging for demonstration cooperatives to implement the principle of democratic management according to the law in order to promote the agricultural cooperatives in the country’s healthy development. Accordingly, the government often assigns a higher weight to democratic management indicators in the evaluation of demonstration cooperatives in an attempt to guide cooperatives to practice the principles of democratic management while also guiding non-demonstration cooperatives to imitate the democratic management practices of demonstration cooperatives and to turn democratic management concepts into actions. Therefore, the first hypothesis of this chapter is proposed:

**H1.** 
*Demonstration cooperatives have a higher level of democratic management.*


According to the inherent logic of imitation proposed in [25], those with high status are more likely to be imitated objects, and the imitators are more likely to imitate these objects. At the same time, the law of logical imitation also emphasizes the asymmetry of imitation behavior among organizations, that is, organizations do not necessarily imitate all behaviors but prefer to imitate those behaviors that can represent the optimal results [31]. As an economic organization, a cooperative’s primary goals are to survive in the market and then, through continuous development and growth, to better serve its members. Economic strength also accounts for the highest proportion of society’s evaluation and expectations of cooperatives [32]. Therefore, economic strength and market competitiveness are the primary external traits of a cooperative and are also the first choices for other cooperatives to imitate. Accordingly, the second hypothesis of this chapter is proposed:

**H2.** 
*Demonstration cooperatives have stronger economic strength.*


The unification of the two identities of members as both owners of cooperatives and users of services (customers) is the fundamental difference between cooperatives and other types of enterprises [33,34]; therefore, serving members is an essential provision and purpose of a cooperative [30]. According to the two central government documents, document No. 5 of 2019 and document No. 8 of 2010, the authors of this paper considered indicators such as the number of members joining cooperatives, helping members increase their income, and driving the number of surrounding farmers representing the cooperative’s serviceability. Previous studies have shown that demonstration cooperatives have more advantages than non-demonstration cooperatives in terms of increasing the income of members [18] and enhancing the social capital of members [35]. The higher the efficiency and performance of demonstration cooperatives in driving members and serving the surrounding farmers, the stronger the potential demonstration and leading role for other surrounding cooperatives. Accordingly, the third hypothesis of this chapter is proposed:

**H3.** 
*Demonstration cooperatives have stronger service capabilities.*


With the upgrading of the consumption structure, consumers’ demands for the quality, safety, and improvement of agricultural products continue to increase, and the production of safe, green, and high-quality agricultural products has become an inevitable trend for agricultural production and operation entities such as cooperatives competing in the market. Previous studies have shown that the vertical integration cooperation model of “company add cooperative add farmers” is an effective development model for the quality and safety management of agricultural products [36]. In terms of the quality control of agricultural products, cooperatives can promote the improvement of the quality and safety of agricultural products to a certain extent [37]. Specifically, they can strengthen moral responsibility, implement standardized production, implement unified purchase and sales, implement brand strategies, and establish traceability systems, among other methods, to control the quality of agricultural products [38]. Other studies have shown that within a cooperative, the higher its demonstration level, the more likely it will become a cooperative with strong food safety service functions [39]. Most of the agricultural products of demonstration cooperatives are aimed at supermarkets or leading enterprises, and the reverse restraint effect of the demand side on the high quality of agricultural products will make demonstration cooperatives pay more attention to the quality and safety of agricultural products [12]. Accordingly, the fourth hypothesis of this chapter is proposed:

**H4.** 
*The product quality of demonstration cooperatives is higher.*


Since the formal implementation of the “Law of the People’s Republic of China on Farmers’ Professional Cooperatives” in 2007, Chinese cooperatives have rapidly developed, and the number of cooperatives has rapidly grown. Some researchers believe that the phenomenon of empty shell cooperatives is particularly serious [40], and many cooperatives take advantage of the state’s preferential policies [11]. There are very few cooperatives in the true sense [41]. At the same time, it should be noted that since the country began the construction of demonstration cooperatives for farmers, a group of demonstration cooperatives with strong economic strength and high social response have come to the fore [42], and they are driving farmers’ income increases [22], cultivating agricultural product brands [43], promoting the improvement of regional agricultural product quality and safety [44], and provoking good social response. Accordingly, the fifth hypothesis of this chapter is proposed:

**H5.** 
*The social responsibility of demonstration cooperatives is better.*


## 3. Data Collection, Models, and Variables

### 3.1. Data Sources

As a major agricultural province in China, Sichuan is also a major province for the development of agricultural cooperatives, and it is representative of the country. By the end of 2019, there were 1.935 million agricultural cooperatives across the country, of which planting accounted for 47% and animal husbandry accounted for 30% (Data source: Department of Rural Cooperative Economic Guidance, Ministry of Agriculture and Rural Affairs). At that time, Sichuan had 104,000 agricultural cooperatives, of which planting accounted for 55% and animal husbandry accounted for 21% (Data source: Department of Agriculture and Rural Affairs of Sichuan Province). The industrial distribution of cooperatives in Sichuan is not much different from the overall level of the country. In terms of quality, by the end of 2021, Sichuan had 106,100 farmer cooperatives, including 528 national demonstration cooperatives, 3178 provincial-level demonstration cooperatives, and 13,168 county-level and above demonstration cooperatives (Data source: Department of Agriculture and Rural Affairs of Sichuan Province), accounting for 12.4%, which was significantly higher than the national average of 7%. Therefore, the authors of this paper used Sichuan as an example, and our data came from a questionnaire survey on farmer cooperatives in 10 counties in Sichuan in July 2021 that was distributed via a stratified sampling method. First, we selected the first five national-level pilot counties in Sichuan to carry out the promotion of the quality improvement of agricultural cooperatives in the whole county: Anzhou District, Luojiang District, Enyang District, Xuzhou District, and Hanyuan County. Second, we selected a non-pilot county next to each pilot county—Jiangyou City, Mianzhu City, Yilong County, Pingshan County, and Tianquan County—for a total of 10 counties as the research area Figure 1. Third, after determining the counties to be investigated, we connected with the agricultural economic stations of the counties. In the cooperative management system, the director of the county’s demonstration cooperatives was called, and in combination with the actual development of the agricultural industry in each county, demonstration cooperatives in the leading agricultural industries in the county were first screened and numbered before we randomly selected 25–30 demonstration cooperatives from the list. Fourth, after the demonstration cooperatives were selected, according to the geographical location of the demonstration cooperative, non-demonstration cooperatives were selected in the same or adjacent villages; if there was no normal operating cooperative in the same or adjacent village, it was abandoned. Finally, after determining the cooperative, we took the chairperson of the cooperative or the manager who is specifically responsible for production and operation as the survey object, and we conducted a questionnaire survey in the form of one-on-one question and answer session. In the end, 516 questionnaires were recovered, 509 valid questionnaires were obtained, and the effective rate of the questionnaire was 98.6%.

It can be seen from Table 1 that most of the sample cooperatives were sponsored by large planting and breeding households, and a considerable part was sponsored by village cadres. From the perspective of leading industries, the sample cooperatives were mainly in the planting industry, accounting for 62%, and the cooperatives in the breeding industry accounted for 36%. From the perspective of industrial scale, most of the sample cooperatives’ industries were medium- and large-scale, indicating that scaled operation has become a development trend of cooperatives. Regarding the demonstration level, the number of demonstration cooperatives in the sample cooperatives was slightly higher than that of non-demonstration cooperatives.

From the perspective of production factor input, the average land size of the sample cooperatives was 41.233 hectares, the average labor employment cost was 379,620 yuan, the average capital investment was 686,620 yuan, and the average training cost was 12,820-yuan (Table 2). In comparison, the land size and other production factor inputs of the demonstration cooperatives were significantly higher than those of the non-demonstration cooperatives.

### 3.2. Model Settings

The main purpose of this paper was to assess the policy effect of the demonstration cooperatives, specifically by estimating the average treatment effect on the treated (ATT) of the demonstration cooperatives. Referring to previous research [45,46,47], ATT is expressed as:(1)$$\begin{array}{ll} ATT & =E\left\{Y_{iS}-Y_{iF}|T_{i}=1\right\}\\ & =E\left\{Y_{iS}|T_{i}=1\right\}-E\left\{Y_{iF}|T_{i}=1\right\} \end{array}$$

In Formula (1), E{•} represents the expectation operator; $Y_{iS}$ and $Y_{iF}$ represent the latent variables of the demonstration cooperatives and non-demonstration cooperatives, respectively; $T_{i}$ represents the processing variable; $T_{i}=1$ represents the demonstration cooperatives; and $T_{i}=0$ represents the non-demonstration cooperatives.

The difficulty in estimating Equation (1) is that the outcome variable $E\left\{Y_{iF}|T_{i}=1\right\}$ of a demonstration cooperative in a non-demonstration cooperative situation cannot be observed. Previous researchers have generally chosen propensity score matching (PSM) to construct a counterfactual framework to solve this problem. The general steps for calculating ATT using PSM are as follows:

The first step is to select the covariates X while trying to include relevant variables that may affect $\left(Y_{iS},Y_{iF}\right)$ and $T_{i}$ satisfy the negligibility assumption [48]. The covariates selected in this paper mainly included three categories: the individual characteristics of the chairperson, the basic characteristics of the cooperative, and the environmental characteristics. The specific variables are shown in Table 1.

The second step is to calculate the propensity score. The authors of paper chose the logit model to estimate the propensity score:(2)$$p\left(X\right)=\mathrm{Pr}\left(T_{i}=1|X\right)=F\left\{h\left(X\right)\right\}=E\left(T_{i}|X\right)$$

The above formula F{•} represents the cumulative density function, and X is the covariate matrix.

The third step is to perform propensity score matching.

(1)The matching method must be selected. There are a variety of matching methods to choose from when using propensity score matching, and there are no obvious differences between the various matching methods. However, due to certain measurement deviations between different matching methods, even if the same sample data are processed, heterogeneous measurement results will be produced. If the results obtained after applying multiple matching methods are similar or even consistent, the matching results are robust and the sample validity is good [49]. Therefore, to enhance the reliability of the results, the authors of this paper selected three mainstream matching methods.
①Radius matching—that is, the absolute distance $\left|p_{i}-p_{j}\right|\leq \varepsilon$ that limits the propensity score ($\varepsilon \leq 0.25{\hat{\sigma }}_{pscore}$), where ${\hat{\sigma }}_{pscore}$ is the sample standard deviation of the propensity score—is generally recommended [49]. After calculation, the authors of this paper set the matching radius to 0.065.
②Second is kernel matching, that is, using the kernel function to calculate the weight w(i,j) [50,51], the weight expression is:(3)$$w\left(i,j\right)=\frac{K\left[\left(x_{j}-x_{i}\right)/h\right]}{\sum_{k:T_{k}=0}K\left[\left(x_{k}-x_{i}\right)/h\right]}$$
where h is the specified bandwidth and K(•) is the kernel function. The authors of this paper used the default kernel function and bandwidth.
③Third is local linear matching, that is, not using kernel regression but using local linear regression to estimate *w(i,j)*.

(2)A balance test must be conducted. If the estimation of the propensity score is accurate, the distribution between the matched treatment group and the control group should be relatively uniform. Generally, the standardized bias is used to test, and the calculation formula is as follows:


(4)
$$\delta =\frac{\left|{\bar{x}}_{treat}-{\bar{x}}_{control}\right|}{\sqrt{\left(s_{x,treat}^{2}+s_{x,control}^{2}\right)/2}}$$


In the above formula, $s_{x,treat}^{2}$ and $s_{x,control}^{2}$ are the sample variances of the treatment group and the control group variable x, respectively, and δ≤10% is generally required. If the standardized deviation is greater than 10%, re-matching is required.

The fourth step is to calculate the average treatment effect. The average treatment effect includes three categories: first the average treatment effect on the treated (ATT), which in this paper is the average value of the changes in the demonstration cooperatives in each indicator. The second is the average treatment effect on the untreated (ATU), that is, the average value of the changes of the non-demonstration cooperatives in each indicator. The third is the average treatment effect (ATE) of the whole sample, that is, the average value of changes in all cooperatives in each indicator. Since this paper was focused on analyzing the policy effect of demonstration cooperatives, we focused on whether the ATT was significant, and its expression is:(5)$$ATT=\frac{1}{N_{1}}\sum_{i:T_{i}=1}\left(y_{i}-{\hat{y}}_{0i}\right)$$
where $N_{1}=\sum_{i}T_{i}$ represents the number of individuals in the treatment group, that is, the number of demonstration cooperatives; $\sum_{i:T_{i}=1}$ represents the sum of the individuals in the treatment group; $y_{i}$ represents the policy effect index value of the demonstration cooperative; and ${\hat{y}}_{0i}$ represents the estimated value of the policy effect index of the demonstration cooperative in the context of not being a demonstration cooperative.

### 3.3. Variable Description

The authors of this paper focused on examining the policy effects of demonstration cooperatives from five respects: democratic management, economic strength, service capability, product quality, and social response. The selection of variables was mainly based on the following considerations.

#### 3.3.1. Explained Variables

(1)The potential policy effect of democratic management. The democratic management of cooperatives includes members’ full right to know, effective participation, equal voting rights, and ultimate control over the decision-making process, especially the distribution plan [12]. To a large extent, members’ understanding and participation in the management and decision making of the cooperative are enacted through the member (representative) general assembly. The establishment, operation, and information disclosure of the board of directors and supervisors are also important ways for members to understand and participate in the cooperative’s affairs. Therefore, two variables, the operation of “three meetings” and the method of surplus distribution, were selected to measure the policy effect of democratic management of the demonstration cooperatives. Among them, the operation of “three meetings” is indicated by whether the cooperative council, supervisory board, and member (representative) meetings are working normally. The method of surplus distribution is expressed by whether the proportion of the cooperative’s distributable surplus returned according to the transaction volume (amount) between the members and the cooperative is not less than 60%.(2)Potential policy effect of economic strength. Four variables were selected to measure the economic strength policy effect of demonstration cooperatives: total investment by members, the fixed assets of cooperatives, the total annual operating income of cooperatives, and the input–output ratio of cooperatives.(3)Potential policy effect of service capability. Four variables were selected to measure the serviceability policy effect of a demonstration cooperative, including the number of members joining the cooperative, the number of annual training people in the cooperative, the average income that members are helping to increase, and the number of surrounding farmers being driven.(4)Potential policy effect on product quality. Two variables, the agricultural product quality certification and the number of registered trademarks, were selected to measure the policy effect on the product quality of demonstration cooperatives.(5)Potential policy effect of social repercussions. Social repercussions mainly refer to the reputation of a cooperative in the local or wider area and the contribution of the cooperative to the local area to obtain corresponding social recognition. Therefore, two variables, the number of times the cooperatives have won commendation awards and the number of jobs created, were selected to measure the policy effect of the demonstration cooperatives’ social response.

#### 3.3.2. Core Explanatory Variables

The authors of this paper focused on whether demonstration cooperatives perform better than ordinary cooperatives in five aspects of democratic management, economic strength, serviceability, product quality, and social response in exerting policy effects. Therefore, whether a cooperative was rated as a demonstration cooperative is the core explanatory variable of this section.

#### 3.3.3. Control Variables

The authors of this paper divided the control variables into three parts: the individual characteristics of the chairperson, the basic characteristics of the cooperative, and the characteristics of the external environment. Specifically, the individual characteristics of the chairperson include the chairperson’s gender, age, education level, management experience, and industrial operation experience. The basic characteristics of the cooperative include the duration of the cooperative, whether it has a fixed office space, and the scale of the industry. The characteristics of the external environment include the distance from the nearest demonstration cooperative in the same industry and the number of affairs supported by the government. Among these, the number of affairs supported by the government was related to 6 items in the questionnaire: e-commerce, financing loan, technology connection, talent training, infrastructure construction, and refrigeration and preservation facility construction. Another open question item “Other” was set up for respondents to fill in. Finally, considering regional differences, the authors of this paper controlled the counties in which cooperatives were located in the model. The variable definitions and descriptive statistics of this paper are shown in Table 3.

It can be seen from Table 4 that in the dimension of democratic management, the operation of the “three meetings” of the demonstration cooperatives was significantly higher than that of the non-demonstration cooperatives at the 1% significance level, and the mean value of surplus distribution method was also significantly higher than that of non-demonstration cooperatives at the 5% significance level; this shows that the demonstration cooperatives have a significant potential policy effect in democratic management. In terms of economic strength, the indicators of total investment, fixed assets, total operating income, and the input–output ratio of demonstration cooperatives were 94.121, 117.649, 161.209, and 0.115 higher, respectively, than those of non-demonstration cooperatives, and both were significant at the 1% significance level. This result intuitively shows that the demonstration cooperatives have a significant potential policy effect in terms of economic strength. At the same time, the eight indicators of the three dimensions of service capability, product quality, and social response of the demonstration cooperatives were also significantly higher than the non-demonstration cooperatives at the 1% significance level. This shows that demonstration cooperatives also have significant potential policy effects in these three dimensions.

## 4. Results and Discussion

### 4.1. Analysis of Influencing Factors of Cooperatives Being Rated as Demonstration Cooperatives

It can be seen from Table 5 that in terms of the personal characteristics, the gender, education level, management experience, and industrial experience of the chairperson had no significant influence on whether a cooperative was rated as a demonstration cooperative. Only the age of the chairperson had a significant positive impact on whether the cooperative was rated as a demonstration cooperative at the 5% significance level. From the perspective of the average marginal effect, for every 1-year increase in the age of the chairperson, the probability of the cooperative being rated as a demonstration cooperative increased by 3.4%. Combined with the coefficient of the square term of the age of the chairperson, it can be seen that the coefficient of the square of the age was negative, indicating that the age of the chairperson was not as high as possible. The reason for this results that an older chairperson generally has higher qualifications, richer life experiences, more experience in dealing with interpersonal relationships with village cadres and government personnel. However, compared with a young chairperson, a too-old chairperson can devote limited energy to the development of their cooperative. Therefore, a too-old chairperson is not conducive to promoting the growth of a cooperative.

Regarding the basic characteristics, the duration of the cooperative, whether the cooperative has a fixed office space, and the industrial scale of the cooperative all had a significant positive impact on the cooperative being rated as a demonstration cooperative at the 1% significance level. Judging from the average marginal effect, the probability of being rated as a demonstration cooperative increased by 2.5% for each additional year of cooperative survival. This is because the longer the survival time, the stronger the survival ability of the cooperative and the corresponding market competitiveness, so the probability of being rated as a demonstration cooperative increases. Consistent with the findings of [52], cooperatives with a longer history were found to perform better than their younger peers in increasing members’ income and identity ratings. Compared with cooperatives without a fixed office space, the probability of a cooperative with a fixed office space being rated as a demonstration cooperative increased by 20.6% because having a fixed office space is an important reference for government departments to select demonstration cooperatives. Both the national demonstration cooperatives and the Sichuan Provincial Demonstration Cooperatives require “fixed office space”. Consistent with the findings of Marcis et al., certifications of good practices in hygiene, health, and safety in the workplace of the cooperatives were observed [53]. Other researchers have also argued that much emphasis has been on building teams through creating trust and loyalty in the workplace [54]. The probability of being rated as a demonstration cooperative was found to increase by 10% for each level of the industrial scale of a cooperative because the larger the industrial scale of a cooperative, the higher the fixed asset investment and industrial operating income and the stronger the economic strength of the cooperative, these qualities are advantageous in the evaluation of demonstration cooperatives. Consistent with the findings of research on Spanish agricultural cooperatives, larger cooperatives can positively affect cooperative performance through competitive advantages such as economies of scale, greater negotiating power, and ease of access to different resources [55].

From the perspective of the environmental characteristics of cooperatives, the relative status of cooperatives and government support were found to have a significant positive impact on whether a cooperative was rated as a demonstration cooperative at the 1% significance level. From the perspective of the average marginal effect, from the cooperative to the nearest demonstration cooperative in the same industry, the probability of being rated as a demonstration cooperative increased by 4.6% for every 1 km increase in the driving distance of a motor vehicle because there is a competitive relationship between cooperatives and demonstration cooperatives in the same industry. The closer the distance, the stronger the competitiveness. In contrast, the competitiveness of demonstration cooperatives was found to be generally stronger than that of ordinary cooperatives. Therefore, the farther away from demonstration cooperatives in the same industry, the more resources that will be obtained through competition and the greater the chance of being rated as a demonstration cooperative. For each additional piece of government support for cooperatives, the probability of a cooperative being rated as a demonstration cooperative increased by 7.7% the more government support cooperatives receive, the more policy resources they obtain and the closer their relationship with grassroots government departments. Therefore, these cooperatives are more likely to be rated as demonstration cooperatives by government departments. Our results are consistent with those of Cox and Le, who argued that a stable legal environment and appropriate government support are extremely important for the successful development of cooperatives [56]. Other researchers have also argued that while most cooperatives are externally initiated, strong state participation and support can help foster continued positive demonstration of cooperatives [57]. Finally, all regional dummy variables were not found to be significant, indicating that there was no significant difference in whether the county cooperatives were rated as demonstration cooperatives.

### 4.2. PSM Matching Results and Common Support Domain Analysis

According to the sample loss results under three different matching methods, the treatment group (demonstration cooperatives) and the control group (non-demonstration cooperatives) lost a total of 26 samples and retained 483 matching samples, indicating that the matching results were better. It can be intuitively understood from Figure 2 that under the three different matching methods, most of the observations were in the common value range (on support) regardless of matching method was used, as only a small number of samples were lost. 

### 4.3. Balance Test Analysis

The results of the balance test in Table 6 show that from the perspective of the standardized deviation changes before and after matching, the standardized deviations of the three methods in the control group and the treatment group after matching were less than 10% except for government support, indicating that the matched data had a good balance. At the same time, radius matching, and kernel matching showed that, except for government support variables, the *t*-test results of all other covariates after matching accepted the null hypothesis that there was no systematic difference between the treatment group and the control group. The results of linear matching showed that the *t*-test results of all covariates after matching accepted the null hypothesis that there was no systematic difference between the treatment group and the control group, and the test results again verified that the matched data were well-balanced.

### 4.4. Analysis of the Policy Effect Results of Demonstration Cooperatives

Table 7 reports the ATTs of the demonstration cooperatives on the five types of policy effect outcome variables calculated with the three matching methods. The results showed that the ATT values calculated by the three methods were consistent in value and significance level, indicating that the sample data had good robustness. Referring to previous research [58] to facilitate the analysis of the empirical results, the authors of this paper took the arithmetic mean of the calculated values of the three matching methods to represent the potential policy effect of the demonstration cooperatives in various respects.

The results of the two variables in democratic management in Table 7 show that the demonstration cooperatives were 5.9% and 1% higher than the non-demonstration cooperatives in terms of the operation of the “three associations” and the surplus distribution method, respectively, but the statistical test results were not significant. These findings show that the policy effect of the demonstration cooperatives in democratic management is not obvious. Further statistical analysis showed that among the 279 sample cooperatives, there were 262 demonstration cooperatives with sound and effective “three associations”, accounting for about 94% of the sample cooperatives. There were only 23 demonstration cooperatives with a transaction volume (amount) rebate ratio of not less than 60%, accounting for only 8% of the sample cooperatives. Therefore, it can be considered that although the demonstration cooperatives have established a perfect “three associations” system, they have not yet established a surplus distribution system that is compatible with democratic management and returned according to the transaction volume (amount), thus making the democratic management of demonstration cooperatives a mere formality. Therefore, in terms of democratic management, it is difficult for a demonstration cooperative to close a gap with other cooperatives, which is consistent with the research conclusions Wang et al. [12]. Only a few present and potential members are aware of democratic institutions in economic and social life, and an indifferent attitude to individual and common needs puts authentic collective action out of reach [59].

In terms of economic strength, the total capital contribution of the members of the demonstration cooperatives was 47.3% higher than that of the non-demonstration cooperatives, and this indicator was significant at the 10% significance level. Combined with the calculation of the mean results in Table 2, it can be seen that after excluding the influence of observable factors, the total contribution of members of the demonstration cooperatives was about 810,000 yuan higher than that of the non-demonstration cooperatives. This shows that the total investment by members of demonstration cooperatives was significantly higher than that of the non-demonstration cooperatives. Some scholars also argue that member capital contributions are linked to product delivery (marketing) rights that attain value and can be transferred, and membership is closed or restricted [60]. The fixed assets of the demonstration cooperatives were found to be 48.6% higher than those of the non-demonstration cooperatives, and this indicator was significant at the 5% significance level. This results showed that the fixed asset investment of the demonstration cooperatives was significantly higher than that of the non-demonstration cooperatives. Another study also demonstrated consolidation and an increase in fixed assets against the background of a decrease in the number of agricultural consumer cooperatives [61]. The total operating income of the demonstration cooperatives was found to be 52.7% higher than that of the non-demonstration cooperatives, and this indicator was significant at the 5% significance level. This shows that from the perspective of total operating income, the policy effect of the demonstration society was significant. The input–output ratio of the demonstration cooperatives was 6.4% higher than that of the non-demonstration cooperatives, but this indicator did not pass the statistical test. Therefore, from the perspective of the input–output ratio, the policy effect of demonstration cooperatives was not shown to be obvious.

According to these results, in terms of economic strength, the demonstration cooperatives have a significant policy effect on the non-demonstration cooperatives in terms of the total index, but in the relative index, the policy effect is not significant. To analyze the reasons in terms of selection criteria, government departments have set very clear threshold conditions for the total amount of cooperative members’ contributions, fixed assets, and annual operating income during the selection process of demonstration cooperatives at all levels, e.g., national demonstration cooperatives require “members to contribute more than 1 million yuan, fixed assets of more than 500,000 yuan in western cooperatives, and annual operating income of more than 1.5 million yuan” and Sichuan provincial demonstration cooperatives require “members to contribute more than 500,000 yuan, cooperatives in hilly areas with fixed assets of more than 300,000 yuan, and annual operating income of more than 800,000 yuan”, and the clear threshold conditions make ideal cooperatives that want to become demonstration cooperatives strive to expand their scale, increase investment in fixed assets, increase total income of cooperatives, and widen the gap with non-demonstration cooperatives in terms of total indicators. However, in terms of relative indicators, the national demonstration cooperative has no stipulations and the Sichuan provincial demonstration cooperative only stipulates that “assets are greater than liabilities, and there has been no continuous loss in the past two years”. The selection criteria do not impose high requirements on the profitability of cooperatives; the cooperatives participating in the evaluation of the demonstration cooperatives only need to ensure that they do not lose money in terms of relative indicators. On the other hand, the natural growth attributes of organisms determine that the input and output of the agricultural industry at the production end are relatively fixed. Therefore, the scale advantage achieved by the demonstration cooperatives can be slightly higher than that of the non-demonstration cooperatives in terms of input and output, but it is difficult to open a relatively obvious gap with the non-demonstration cooperatives.

In terms of service capacity, the number of members of the demonstration cooperatives was found to be 57.8% higher than that of the non-demonstration cooperatives, and this indicator was significant at the 1% significance level, indicating that the number of members of the demonstration cooperatives was significantly higher than the non-demonstration cooperatives. The annual training number of the demonstration cooperatives was 73.2% higher than that of the non-demonstration cooperatives, and this indicator was significant at the 1% significance level, showing that demonstration cooperatives have a very significant policy effect on non-demonstration cooperatives in organizing members and non-member farmers to participate in technical training. The demonstration cooperatives could receive favorable support and services from the government, including free training in quality control and product certification [62]. Demonstration cooperatives were found to help member farmers increase their income by 9.4% more than non-demonstration cooperatives, and this indicator was significant at the 1% significance level, indicating that the demonstration cooperatives have a very significant policy effect on non-demonstration cooperatives in promoting the income increase of their members. The number of surrounding farmers driven by the demonstration cooperatives was found to be 64.0% higher than that of the non-demonstration cooperatives, and this indicator was significant at the 1% significance level, indicating that the demonstration cooperatives have a very significant policy effect on the non-demonstration cooperatives in terms of radiating and driving surrounding farmers. The authors of other studies have reached similar conclusions in terms of the service capacity of demonstration cooperatives. With increasing demonstration levels, the ability of agricultural cooperatives to provide marketing services to their members tends to increase as well [63]. Cooperative organizations are expected to provide an appropriate avenue for the demonstration of modern technologies to meet farmers’ needs in agricultural production and processing [64].

In terms of product quality, that of the demonstration cooperatives was found to be 28.7% higher than that of the non-demonstration cooperatives, and this indicator was significant at the 5% significance level, indicating that the demonstration cooperatives have a more significant role in improving the quality of agricultural products than the non-demonstration cooperatives. The number of registered trademarks owned by the demonstration cooperatives was 26.3% higher than that of the non-demonstration cooperatives, and this indicator was significant at the 5% significance level, indicating that the demonstration cooperatives have a significant policy effect on the non-demonstration cooperatives in terms of product brand building.

In terms of social repercussions, the number of awards awarded by demonstration cooperatives was found to be 1.4 times that of non-demonstration cooperatives, and this indicator was significant at the 1% significance level, indicating that the social recognition degree of demonstration cooperatives is higher than that of the non-demonstration cooperative. The number of jobs created by the demonstration cooperatives was 82.2% higher than that of the non-demonstration cooperatives, and this indicator was significant at the 10% significance level, indicating that the demonstration cooperatives have a certain policy effect on the non-demonstration cooperatives in promoting employment.

### 4.5. Robustness Check

Although the results of PSM estimation are robust, PSM is limited to controlling the selection bias of observable variables and cannot address the selection bias caused by unobservable factors. Therefore, to further test the robustness of the estimated results and avoid unobservable factors affecting the results in the PSM estimation process, we drew on previous research [12] and used inverse probability weighting-regression adjustment (IPWRA) to test the robustness.

The assumption of the PSM method in the treatment effect estimation is that if the treatment variable equation is correctly set, the estimated results will be consistent and unbiased; however, if the outcome equation is incorrectly set in the PSM model, the estimated treatment effect will be biased and inconsistent. In contrast, the IPWRA method is doubly robust, that is, if either the choice equation or the result equation is correctly set, a consistent estimate can be made [12]. Table 8 reports the robustness test results of the IPWRA method.

It can be seen from Table 6 that, except for the operation of the “Three Meetings”, fixed assets, input–output ratio, and product quality certification, the treatment effects and their significance levels of the remaining outcome variables were not significantly different from the PSM estimation results, indicating that our research had good robustness.

## 5. Conclusions and Policy Recommendations

Global policymakers see cooperatives as an institutional tool for agricultural development. The Chinese government has also vigorously developed agricultural cooperatives since 2007 to improve the level of organization of farmers and to promote the development of modern agriculture in the country. Since 2009, the Chinese government has implemented large-scale demonstration cooperative construction actions to improve the overall development quality of cooperatives. However, though China is largest developing country in the world, there no empirical study has been conducted to explore the role and potential scope of demonstration cooperative construction. The authors of this paper measured the policy evaluation of the demonstration cooperative construction in China and provide policy recommendations to promote the further development of cooperatives. Based on the nudge–imitation theory, the authors of this paper used the PSM model to empirically analyze whether the demonstration cooperatives performed better than non-demonstration cooperatives in the five respects of democratic management, economic strength, service capability, product quality, and social response expected by the government. Furthermore, the IPWRA method was used to test the robustness of the empirical results found with the PSM method.

The results of this paper showed that the demonstration cooperatives are significantly better than the non-demonstration cooperatives in the four aspects of economic strength, service ability, product quality, and social response, but the development difference between the demonstration cooperatives and the non-demonstration cooperatives in terms of democratic management was not found to be significant. Therefore, the demonstration cooperatives have significant policy effects in four aspects: improving the economic strength of cooperative, enhancing the service capacity of cooperatives, improving the quality of cooperatives’ products, and expanding the social response of cooperatives. However, the government’s construction of demonstration cooperatives has not played the expected demonstration role in improving the level of the democratic management of cooperatives. Therefore, the future development of China’s farmer cooperatives and demonstration cooperatives is predictable in at least the following two aspects. Firstly, because the construction of demonstration cooperatives has significant policy effects in promoting the economic strength of cooperatives, the Chinese government will continue to strengthen the construction of demonstration cooperatives and invest more policy resources into the demonstration cooperatives. Against the background of policies such as improving the quality of cooperatives and promoting the construction of the whole county, the development quantity and quality of the demonstration cooperatives of Chinese farmers’ cooperatives are likely to enter a new stage of development in the future. Secondly, since the current policy effect of the demonstration cooperatives in democratic management is not significant, improving the democratic management level of cooperatives is an important challenge that China will face in improving the internal governance of agricultural cooperatives.

An important policy implication is that the Chinese government should continue to strengthen policy support for demonstration cooperatives because the existing policy support has achieved good results in promoting economic development, serving farmers, and improving the product quality of demonstration cooperatives. At the same time, the government should also provide more effective measures to promote the democratic management of demonstration cooperatives, e.g., by strengthening the democratic education of cooperative members and linking the normative level of internal governance of the cooperative with the support of policy funds to guide cooperatives to improve the level of governance.

Some limitations of our analysis should be kept in mind. First, we only collected data from a sample survey in Sichuan Province and lacked a comprehensive survey of the development of cooperatives across the country. Therefore, it remains to be seen whether the research conclusions can be extended to other provinces and cities in China. Second, we only studied whether demonstration cooperatives are better than non-demonstration cooperatives and did not further analyze whether the construction of demonstration cooperatives has led to the development of non-demonstration cooperatives. As such, our analysis only verifies that demonstration cooperatives drive other cooperatives; it cannot verify the final result of the demonstration.

## Figures and Tables

**Figure 1 ijerph-19-12259-f001:**
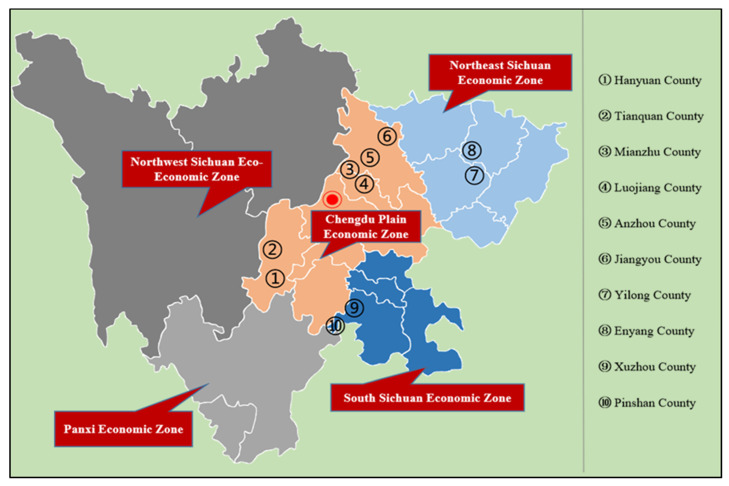
Geographical distribution of sample counties in Sichuan Province.

**Figure 2 ijerph-19-12259-f002:**
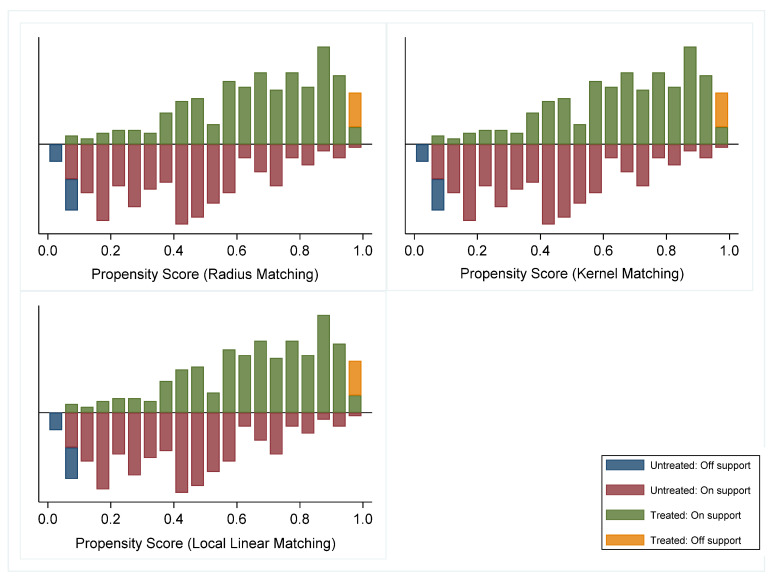
Common support areas for propensity scores.

**Table 1 ijerph-19-12259-t001:** Basic statistics of sample cooperatives.

		Frequency	Proportion (%)			Frequency	Proportion (%)
Type of cooperative	Big planters lead	313	61.49	Leading industry	Crop farming	316	62.08
	Village cadres lead	91	17.88		Animal husbandry	181	35.56
	Company lead	45	8.84		Service industry	12	2.36
	Other	60	11.79				
Industrial scale	out of scale	40	7.86	Demonstration level	Demonstration cooperative	279	54.81
	small scale	70	13.75		Of which: county and city level	183	35.95
	medium scale	215	42.24		Provincial and National	96	18.86
	Large scale	184	36.15		Non-demonstration cooperative	230	45.19

Note: The classification criteria for industry scale are as follows, according to the statistical yearbook and the standards issued by Sichuan Province: the planting category is divided into 30 mu, 100 mu, and 500 mu; hog breeding is divided into 30, 100, and 1000 heads; cattle and sheep breeding are divided into 50, 100, and 300 heads; and the poultry category is divided into 300, 1000 and 10,000 birds.

**Table 2 ijerph-19-12259-t002:** Comparison of the mean difference between the input of production factors of demonstration cooperatives and non-demonstration cooperatives.

	Total Sample	Demonstration Cooperatives	Non-Demonstration Cooperatives	Mean Diff.
Land size	41.233	50.673	29.783	20.980 *
Labor cost	37.962	49.652	23.782	25.870 ***
Capital investment	68.662	93.973	37.958	56.015 ***
Training cost	1.282	1.573	0.930	0.643 ***

Note: * and *** indicate significance at the 10% and 1% significance levels, respectively.

**Table 3 ijerph-19-12259-t003:** Variable definitions and descriptive statistics.

Variable Name	Variable Definitions	Mean	S.D.
**Explained variable**			
Operation of “three meetings”	The directors’ board, the supervisors’ board, and the members’ general assembly (representatives) are sound and effectively functioning; yes = 1, no = 0	0.837	0.369
Surplus distribution method	The ratio of distributable surplus to be returned according to the trading volume (amount) between members and the club shall not be less than 60%; yes = 1, no = 0	0.063	0.243
Total Membership Contributions	The actual total investment of cooperative members (10 thousand yuan)	130.24	201.75
Fixed assets	The total fixed assets of cooperatives (ten thousand yuan)	153.41	223.28
Total operating income	The total income of the cooperative in 2020 (ten thousand yuan)	162.45	353.25
Input–output ratio	The ratio of total investment to the total income of cooperatives in 2020	0.768	0.359
Number of Members	The actual number of members of the co-op as of the end of 2020	68.141	107.9
Annual training	The total number of people organized by the cooperative in 2020 for members and non-member farmers to conduct intensive training	97.922	220.50
Help members increase their income	In 2020, whether the cooperative helped members to increase their average income. Based on the per capita disposable income of rural residents, residents, and urban residents in Sichuan Province in 2020 (1.6, 2.7, and 3.8 ten thousand yuan, respectively), it was divided into four intervals: [0, 1.6) = 1, [1.6, 2.7) = 2, [2.7, 3.8) = 3. [3.8, +∞) = 4	1.063	0.307
Drive the number of farmers	Number of surrounding farmers driven by cooperatives by the end of 2020	147.60	287.84
Product quality certification	Product quality certification level owned by the cooperative: no certification = 0; pollution-free product certification = 1; green food certification = 2; organic food certification = 3	0.727	1.147
Number of registered trademarks	Total number of registered trademarks owned by cooperatives	0.473	0.994
Number of awards	The total number of praises, awards, and honorary titles received by cooperatives	1.573	3.435
Number of jobs	Number of permanent jobs that cooperatives can provide	4.055	4.159
**Core explanatory variables**			
Demonstration cooperative	Whether the cooperative is a demonstration cooperative; yes = 1, no = 0	0.549	0.498
**Select Equation Control Variables**			
**Chairperson characteristics**			
Gender	Male = 1, Female = 0	0.820	0.385
Age	Chairperson’s age	46.192	8.958
Education	Education years of the chairperson	11.347	3.241
Management experience	The length of the chairperson’s management experience (years)	10.169	7.635
Industry experience	The length of time that the chairperson has been engaged in the current industrial management (years)	8.331	6.638
**Cooperative characteristics**			
Duration	Co-op survival time by the end of 2020 (years)	5.769	3.058
Workplace	Does the cooperative have a permanent office; yes = 1, no = 0	0.88	0.315
Industrial scale	Scattered planting, free-range farming = 1; small scale = 2; medium scale = 3; large scale = 4	3.069	0.900
**Environmental characteristics**			
Relative position	Motor vehicle driving distance from the cooperative to the nearest demonstration cooperative of the same industry (km)	9.641	15.707
Governmental support	Number of government-supported affairs for cooperatives in 2020	1.075	1.037

**Table 4 ijerph-19-12259-t004:** Comparison of mean differences of explained variables between demonstration cooperatives and non-demonstration cooperatives.

Variables	Demonstration Cooperatives	Non-Demonstration Cooperatives	Mean Diff.
Operation of “three meetings”	0.939	0.713	0.226 ***
Surplus distribution method	0.082	0.039	0.043 **
Total Membership Contributions	173.033	78.912	94.121 ***
Fixed assets	206.882	89.233	117.649 ***
Total operating income	235.427	74.217	161.209 ***
Input–output ratio	0.820	0.705	0.115 ***
Number of Members	95.398	35.348	60.050 ***
Annual training	144.685	41.622	103.063 ***
Help members increase their income	1.100	1.017	0.083 ***
Drive the number of farmers	205.548	77.526	128.022 ***
Product quality certification	0.954	0.452	0.501 ***
Number of registered trademarks	0.682	0.217	0.465 ***
Number of awards	2.461	0.491	1.969 ***
Number of jobs	4.893	3.035	1.858 ***

Note: ** and *** indicate significance at the 5% and 1% significance levels, respectively.

**Table 5 ijerph-19-12259-t005:** Results of selection model for whether cooperatives become demonstration cooperatives.

Variables	Coefficients (Std. Error)		Marginal Effects (Std. Error)	
Gender	−0.105	(0.293)	−0.019	(0.052)
Age	0.188 **	(0.096)	0.034 **	(0.017)
Age squared	−0.002	(0.001)	0.000	(0.000)
Education	0.004	(0.037)	0.001	(0.007)
Management experience	−0.026	(0.016)	−0.005	(0.003)
Industry experience	0.027	(0.022)	0.005	(0.004)
Duration	0.140 ***	(0.048)	0.025 ***	(0.008)
Workplace	1.154 ***	(0.387)	0.206 ***	(0.067)
Industrial scale	0.558 ***	(0.141)	0.100 ***	(0.024)
Relative position	0.260 ***	(0.099)	0.046 ***	(0.017)
Governmental support	0.433 ***	(0.117)	0.077 ***	(0.020)
Hanyuan county	1.178	(0.832)		
Jiangyou county	0.254	(0.739)		
Luojiang county	0.583	(0.716)		
Mianzhu county	−0.873	(0.726)		
Anzhou county	0.875	(0.796)		
Pingshan county	−0.052	(0.738)		
Xuzhou county	0.271	(0.723)		
Yilong county	1.244	(0.782)		
Enyang county	−0.05	(0.628)		
Constants	−9.437 ***	(2.523)		
Observations	509			
Wald chi^2^ (20)	99.65 ***			
Pseudo R^2^	0.229			
Log pseudolikelihood	−270.319			

Note: Values in brackets are standard errors; ** and *** indicate significance at the 5%, and 1% significance levels, respectively.

**Table 6 ijerph-19-12259-t006:** Matching balance test results.

Covariates	Unmatched Matched	Radius Matching		Kernel Matching		Local Linear Regression Matching	
		%bias	*t*-Test	%bias	*t*-Test	%bias	*t*-Test
Gender	Unmatched	1.00	0.12	1.00	0.12	1.00	0.12
	Matched	8.80	1.00	8.00	0.91	7.00	0.81
Age	Unmatched	24.20	2.74 ***	24.20	2.74 ***	24.20	2.74 ***
	Matched	−6.70	−0.76	−4.80	−0.55	−7.90	−0.91
Education	Unmatched	12.80	1.45	12.80	1.45	12.80	1.45
	Matched	5.00	0.59	3.90	0.46	3.00	0.37
Management experience	Unmatched	22.80	2.56 **	22.80	2.56 **	22.80	2.56 *
	Matched	−1.00	−0.11	−1.30	−0.14	−5.70	−0.62
Industry experience	Unmatched	36.40	4.09 ***	36.40	4.09 ***	36.40	4.09 ***
	Matched	−10.30	−0.96	−6.50	−0.62	−11.90	−1.10
Duration	Unmatched	64.90	7.19 ***	64.90	7.19 ***	64.90	7.19 ***
	Matched	6.50	0.70	8.50	0.92	−1.00	−0.10
Workplace	Unmatched	51.00	5.91 ***	51.00	5.91 ***	51.00	5.91 ***
	Matched	−1.20	−0.23	−1.20	−0.23	1.50	0.27
Industrial scale	Unmatched	76.20	8.65 ***	76.20	8.65 ***	76.20	8.65 ***
	Matched	9.60	1.26	10.00	1.32	8.90	1.15
Relative position	Unmatched	36.80	4.14 ***	36.80	4.14 ***	36.80	4.14 ***
	Matched	−7.50	−0.87	−5.20	−0.61	−5.30	−0.63
Governmental support	Unmatched	51.30	5.67 ***	51.30	5.67 ***	51.30	5.67 ***
	Matched	15.50	1.71 *	15.20	1.68 *	14.40	1.59
Hanyuan county	Unmatched	23.70	2.6 **	23.70	2.6 **	23.70	2.6 **
	Matched	−8.60	−0.78	−5.70	−0.53	−12.40	−1.10
Jiangyou county	Unmatched	11.30	1.26	11.30	1.26	11.30	1.26
	Matched	−5.20	−0.52	−5.70	−0.57	−7.10	−0.70
Luojiang county	Unmatched	19.60	2.17 **	19.60	2.17 **	19.60	2.17 **
	Matched	6.90	0.77	6.10	0.67	4.20	0.46
Mianzhu county	Unmatched	−4.70	−0.54	−4.70	−0.54	−4.70	−0.54
	Matched	3.50	0.44	1.60	0.19	2.20	0.27
Anzhou county	Unmatched	19.80	2.18 **	19.80	2.18 **	19.80	2.18 **
	Matched	4.50	0.48	5.60	0.61	3.70	0.40
Pingshan county	Unmatched	4.70	0.53	4.70	0.53	4.70	0.53
	Matched	0.60	0.07	−0.10	−0.01	2.90	0.33
Xuzhou county	Unmatched	9.30	1.03	9.30	1.03	9.30	1.03
	Matched	−6.70	−0.69	−8.80	−0.89	−9.50	−0.96
Yilong county	Unmatched	24.90	2.73 ***	24.90	2.73 ***	24.90	2.73 ***
	Matched	−5.50	−0.52	−1.10	−0.11	−3.40	−0.33
Enyang county	Unmatched	−55.40	−6.24 ***	−55.40	−6.24 ***	−55.40	−6.24 ***
	Matched	2.90	0.34	2.50	0.29	6.90	0.82

Note: *, ** and *** indicate significance at the 10%, 5%, and 1% significance levels, respectively.

**Table 7 ijerph-19-12259-t007:** Potential policy effect results of demonstration cooperatives.

Variables	Radius Matching		Kernel Matching		Local Linear Regression Matching		Mean of ATT
	ATT (S.D.)	*t*-Stat.	ATT (S.D.)	*t*-Stat.	ATT (S.D.)	*t*-Stat.	
Operation of “three meetings”	0.056 (0.052)	1.08	0.061 (0.050)	1.23	0.059 (0.054)	1.10	0.059
Surplus distribution method	0.009 (0.028)	0.32	0.012 (0.027)	0.43	0.008 (0.029)	0.27	0.010
Total Membership Contributions	0.456 * (0.272)	1.67	0.499 * (0.262)	1.90	0.463 * (0.281)	1.65	0.473
Fixed assets	0.484 ** (0.225)	2.15	0.530 ** (0.217)	2.44	0.443 * (0.232)	1.91	0.486
Total operating income	0.530 ** (0.223)	2.38	0.560 *** (0.214)	2.61	0.490 ** (0.230)	2.13	0.527
Input–output ratio	0.060 (0.046)	1.29	0.068 (0.044)	1.54	0.064 (0.048)	1.34	0.064
Number of members	0.596 *** (0.168)	3.55	0.594 *** (0.162)	3.67	0.545 *** (0.174)	3.14	0.578
Annual training	0.745 *** (0.242)	3.08	0.767 *** (0.233)	3.29	0.684 *** (0.250)	2.74	0.732
Help members increase their income	0.094 *** (0.028)	3.38	0.093 *** (0.027)	3.41	0.094 *** (0.028)	3.38	0.094
Drive the number of farmers	0.665 *** (0.210)	3.16	0.648 *** (0.202)	3.21	0.606 *** (0.217)	2.79	0.640
Product quality certification	0.288 ** (0.134)	2.15	0.301 ** (0.129)	2.32	0.271 ** (0.138)	1.97	0.287
Number of registered trademarks	0.253 ** (0.105)	2.42	0.268 *** (0.102)	2.63	0.268 ** (0.107)	2.51	0.263
Number of awards	1.419 *** (0.277)	5.12	1.422 *** (0.274)	5.19	1.384 *** (0.280)	4.94	1.408
Number of jobs	0.828 * (0.428)	1.93	0.833 ** (0.417)	2.00	0.806 * (0.438)	1.84	0.822

Note: Values in brackets are standard errors; *, **, and *** indicate significance at the 10%, 5%, and 1% significance levels, respectively.

**Table 8 ijerph-19-12259-t008:** IPWRA robustness test results.

Variables	ATT	S.E.	Z-Stat.
Operation of “three meetings”	0.038 *	0.023	1.688
Surplus distribution method	0.017	0.030	0.556
Total Membership Contributions	0.356 *	0.183	1.948
Fixed assets	0.233	0.167	1.393
Total operating income	0.462 **	0.182	2.539
Input–output ratio	0.054 *	0.032	1.688
Number of members	0.524 ***	0.117	4.467
Annual training	0.579 ***	0.170	3.395
Help members increase their income	0.091 ***	0.024	3.747
Drive the number of farmers	0.640 ***	0.155	4.143
Product quality certification	0.187	0.120	1.555
Number of registered trademarks	0.305 ***	0.096	3.165
Number of awards	1.524 ***	0.288	5.292
Number of jobs	0.901 **	0.416	2.169

Note: *, ** and *** indicate significance at the 10%, 5%, and 1% significance levels, respectively.

## Data Availability

Not applicable.

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
