# Peer review of "Policy Evaluation of Demonstration Cooperative Construction: Evidence from Sichuan Province, China"

_ijerph, 2022, doi:10.3390/ijerph191912259_

Round 1

Reviewer 1 Report

Useful research with a high potential and meaning for policy makers to assess whether current support policies and regulations promote and enable development of cooperatives in China. For the international scientific readership, the paper can be useful for comparative studies.

I recommend to exclude number from the title: it is not necessary to mention that evidence was gathered from 509 cooperatives. You explain this later in the methodology part. Abstract comprises all relevant information; however, some sentences should be placed in other sequence for more logical abstract structure. I suggest that all text in the abstract follows this logic: 1) describe the problem of the research and its aim, 2) describe the methodology (thus the final sentence of the abstract should be here), 3) mention the main results, and 4) point out some benefits of the results for  the readers/ further studies. 

The aim of the research must be included in the introduction. In the current manuscript it can be found only in in the 3.2. section. I also suggest to use more gender-neutral expressions, for example, instead of “chairman” (line 334 and forward) it is better to use “chairperson”. Some information about the ethical guidelines during the research must be included in the methodology.

The authors have referred to many studies, but I would suggest to include more international experience regarding cooperative analysis.

Author Response

Thank you we have revsed the manuscript accordingly.

Reviewer 2 Report

Dear Authors,

The manuscript presents a quite interesting study, but I have some observation:

-       In medium how large is such a demonstration cooperative, in terms of members, land, etc?

-       What do you mean by”democratic management”, line 571?

-       At line 590 you state that ” In terms of economic strength, the total capital contribution of the members of the demonstration cooperatives is 47.3% higher than that of the non-demonstration cooperatives, and this indicator is significant at the 10% significance level.” What ist in medium the total capital contribution of the members of a demonstration cooperative?

-       It will be important so see you own proposals for the future in terms of demonstration cooperative!

Author Response

Thank you we have revised the manuscript accordingly.
